# Preparation of Transparent Fast-Growing Poplar Veneers with a Superior Optical Performance, Excellent Mechanical Properties, and Thermal Insulation by Acetylation Modification Using a Green Catalyst

**DOI:** 10.3390/polym14020257

**Published:** 2022-01-08

**Authors:** Wen He, Rui Wang, Feiyu Guo, Jizhou Cao, Zhihao Guo, Han Qiang, Shuang Liang, Qunyan Pang, Bairen Wei

**Affiliations:** 1College of Materials Science and Engineering, Nanjing Forestry University, Nanjing 210037, China; ruiwang0909@163.com (R.W.); gfy1558@163.com (F.G.); cjz6127@163.com (J.C.); gzh3398@163.com (Z.G.); qianghan1016@163.com (H.Q.); liangshuang_1998@126.com (S.L.); p15156046514@163.com (Q.P.); Wbr1828075882@163.com (B.W.); 2Innovation Center of Efficient Processing and Utilization of Forest Resources, Nanjing Forestry University, Nanjing 210037, China

**Keywords:** fast-growing poplar veneers, acetylated modification, interface compatibility, optical performance, thermal conductivity

## Abstract

There has been growing interest in transparent conductive substrates due to the prevailing flexible electron devices and the need for sustainable resources. In this study, we demonstrated a transparent fast-growing poplar veneers prepared by acetylated modification, followed by the infiltration of epoxy resin. The work mainly focused on the effect of acetylation treatment using a green catalyst of 4-Dimethylpyridine on the interface of the bulk fast-growing poplar veneer, and the result indicated that the interface hydrophobicity was greatly enhanced due to the higher substitute of acetyl groups; therefore, the interface compatibility between the cell wall and epoxy resin was improved. The obtained transparent fast-growing poplar veneers, hereafter referred to as TADPV, displayed a superior optical performance and flexibility, in which the light transmittance and haze were 90% and 70% at a wavelength of 550 nm, respectively, and the bending radius and bending angle parallel to grain of TADPV were 2 mm and 130°, respectively. Moreover, the tensile strength and tensile modulus of the TADPV were around 102 MPa and 198 MPa, respectively, which is significantly better than those of the plastic substrates used in flexible electron devices. At the same time, the thermal conductivity tests indicated that TADPV has a low coefficient of thermal conductivity of 0.34 Wm^−1^ K^−1^, which can completely meet the needs of transparent conductive substrates. Therefore, the obtained TADPV can be used as a candidate for a flexible transparent substrate of electron devices.

## 1. Introduction

With the development of electronics technology, the demand for electron devices with excellent optical performance, portability, and flexibility, as well as being environment-friendly, has increased. Recently, a lot of research has been focused on the biomass materials to replace plastic substrates due to its degradability, sustainability, easy-processing, and being environment-friendly [1]. Nowadays, nanocellulose films are widely reported as conductive substrates because of their high transparency and tensile strength; however, the large energy consumption, complicated process, and high cost have greatly restricted their further application [2]. Thankfully, the emergence of transparent wood has substantially overcome these disadvantages, as it still retains the advantages of a low density, high transparency, and toughness, as well as biodegradability [3].

Compared to natural wood, the dimensional stability and flexibility of transparent wood are greatly improved due to the impregnation of polymers. As a porous hydrophilic biomaterial, the physicochemical properties of wood are greatly affected by the change of environmental moisture and temperature; however, the transparent polymers used commonly, such as epoxy resin [4], polyving akohol [5], and polystyrene [6], are nearly absolutely hydrophobic materials; therefore, the incompatibility of the interface results in a great deal of micro-nano gaps among the wood cell wall and polymers. On the other hand, the repeated shrinkage and expansion of the cell wall would cause an increase in local stress because of the constant changing of environmental temperature and humidity, which would weaken the interface connection and lead to massive interface separation [7]. Accordingly, the interface separation further increases the interspace in the interior of transparent wood, which brings about a great deal of light scattering and hence greatly decreases the light transmittance of wood. At the same time, the existing of large number of gaps in transparent wood would also result in a decrease in mechanical performance [8]. Therefore, it is very necessary to improve the interface compatibility between the cell wall of wood and transparent polymers.

Among the methods of interface modification of wood, acetylation treatment with acetic anhydride is recognized as one of the most effective ways, in which the reagents are induced to react at the cell wall level and a great deal of covalent bonds are formed between the hydroxyls of wood and acetyl groups by esterification reactions. Kyle et al. [9] modified delignified wood with acetic anhydride and triethylamine as a catalyst. Chen et al. [10] reported the effect of acetylation modification on the transparent wood with different thicknesses using pyridine as a catalyst. Motanari et al. [11] treated Balsa veneers using 2,2-azodiisobutyronitrile, and the varied acid anhydrides to prepare transparent wood. These results fully prove that the transparency was enhanced due to the improvement of the interaction between wood substrates and transparent resins. However, most of catalysts used in acetylation are strong acid or organic compounds, which likely lead to partial hydrolysis for wood materials and therefore bring about a decrease in mechanical performance. Organic compounds, such as pyridine, are generally irritant and deleteriousness, which is harmful to human health. As a result, environment-friendly and mild acidic catalysts have attracted a lot of attention when applied in the acetylation of biomass materials. Avila Ramirez et al. [12] successfully introduced acetyl into bacterial cellulose with citric acid as a catalyst; nevertheless, when used in the bulky wood for acetylation modification, the catalyzing reaction was extremely slow and resulted in a low substitution degree. 4-Dimethylpyridine (DMAP), as a green catalyst, is widely applied in the processes of esterification, etherification, and alkylation, and exhibits a significant catalytic effect [13]. Recently, DMAP was also used in the catalytic modification of different biomass-based celluloses, and achieved remarkable results. However, DMAP has not properly been applied in acetylation modification for bulky wood, and its catalytic effect on large size wood needs to be studied [14].

Therefore, in this study, in order to prepare transparent fast-growing poplar veneers with an outstanding optical performance and high tensile strength, as well as superior flexibility, the poplar veneers are firstly delignified using an acidic sodium chlorite solution, and then are subjected to acetylation treatment with acetic anhydride and DMAP as the catalyst by regulating the reaction time. The cell wall after acetylation modification is investigated systematically [15]. Subsequently, the delignified poplar veneer is infiltrated with epoxy resin by vacuum impregnation, and then followed with low-temperature curing. As a result, the effect of the acetylation process on the optical properties, mechanical strength, and thermal conductivity of the poplar veneers are comprehensively estimated.

## 2. Experimental

### 2.1. Materials

The rotary-cutting poplar veneers, acquired from the sapwood of Nan Kangyang (Populus), with a density of 0.429 g/cm^3^ and a moisture content of 12%, as well as a thickness of 0.5 mm, were obtained from Shengyu Wood Industry Co., Ltd., located in Siyang county of Jiangsu province (Suqian, China). Sodium chlorite (NaClO_2_), acetic acid (CH_3_COOH), and acetic anhydride (C_4_H_6_O_3_) were purchased from Shanghai Maclin Biochemical Technology Co., Ltd. (Shanghai, China). 4-Dimethylpyridine(C_7_H_10_N_2_), hydrogen peroxide, and hydrochloric acid, as well as sodium hydroxide, were obtained from Sinopharm Chemical Reagent Co., Ltd. (Shanghai, China). Epoxy resin AB glue (JH-5511) was obtained from Hangzhou Sihui Port Adhesive Co., Ltd. (Hangzhou, China). The deionized water came from our laboratory, and all chemical reagents were of analytical grade.

### 2.2. Preparation of Transparent Poplar Veneers

The rotary-cutting poplar veneer (PV) was tailored into uniform specimens with dimensions of 5 × 5, and all PVs were dried at 60 °C for 24 h using an oven. Subsequently, the PVs were delignified in an acidic NaClO_2_ solution (5 wt.%) at 75 °C and were stirred continuously with a magnetic stirrer, and the solution was alternated with the fresh NaClO_2_ solution every hour, in which the mass ratio of the PV and NaClO_2_ solution was 1:60. This process was repeated eight times, followed by washing in a mixed solution of ethanol and deionized water (mass ratio was 1:1) until the pH was around 7.

The delignified PV (DPV) was vacuum pumped for 15 min and was placed into an acetic anhydride solution, in which the mass ration was 1:40. Next, 4-Dimethylpyridine was added into the above solution and the acetylation reaction was carried out in an oil bath at 60 °C for 1, 3, and 5 h, respectively. Hereon, the acetylated DPVs are abbreviated as ADPV_1_, ADPV_3_, and ADPV_5_, respectively. After the acetylation reaction, all ADPVs were thoroughly cleansed in an ethanol solution with a rocking sieve, followed by freeze drying for further use. All ADPVs were impregnated with epoxy resin under pumping vacuum for 5 min, and were then taken out and placed in an atmospheric environment for 2 min. This process was repeated three times and the superfluous epoxy resin was removed from the surface of the ADPVs with a razor blade. Finally, the ADPVs filled with epoxy resin were cured at 60 °C for 2 h; accordingly, the cured ADPVs were abbreviated as TADPV_1_, TADPV_3_, and TADPV_5_, respectively. The preparation process of the transparent poplar veneers is shown in Figure 1.

### 2.3. Characterization

The microstructures of all specimens were observed by scanning electron microscopy (SEM) (S4800F, JEOL, Tokyo, Japan), and the change of functional groups in the specimens was determined using FT-IR (VERTEX 80 V, Bruker, Germany). The structure of all specimens was checked by X-ray diffraction (XRD) using Cu-K radiation (Ultima IV, Rigaku, Japan).

The hydrophobicity of ADPVs was evaluated with a contact angle meter (SL200KB, Stockholm, Sweden). The transmittance and haze of the TADPVs were measured using a UV visible spectrophotometer (UV−VIS, Lambda 950, PE, Seattle, WA, USA) according to “ASTM D1003 Standard Method for Haze and Luminous Transmittance of Transparent Plastics”. The mechanical property of 50 × 5 mm TADPVs was tested with a microcomputer control electronic universal testing machine (SANS 4304, MTS, Olympia, WA, USA). The thermal conductivities were measured using a steady state laser-infrared camera thermal conductivity characterization system.

## 3. Results and Discussion

### 3.1. Morphological Analysis

As a biomass material with a multiscale pore structure, wood mainly consists of cellulose, hemicellulose, and lignin. Generally, natural wood is mostly brown and opaque in the visible light range, which is mainly attributed to the strong absorption of lignin on light and the light scattering caused by its uneven structure [16]. In order to prepare a transparent wood, it is firstly necessary to remove the lignin from the wood. As shown in Figure 2a,d, it is clear that the color of the poplar veneer (PV) changed from yellow-brown to pure white after delignification treatment. The fundamental tissue of the PV was arranged densely, and the cell wall was intact, and the cell tissues were tightly bound through the intercellular layer (Figure 2b). Typically, lignin of fast-growing poplar is mainly located at the secondary wall of the cell tissues, the cellular corner, and compound intercellular layer of cells [17]. Obviously, the intercellular layer was completely damaged, and the substances existing in the cell corner were removed entirely; at the same time, the cell cavity was greatly largened and the cell wall became thinner after delignification (Figure 2e). Moreover, the density of PV was significantly decreased from 0.429 g/cm^3^ to 0.334 g/cm^3^ due to the removal of lignin (Figure 2c). The decrease was around 22.06%, which is close to the lignin content of 22.69% for fast-growing polar [18]. This suggests that the lignin in PV was substantially removed.

Subsequently, the microstructure of the cell wall for the delignified PV (DPV) after acetylation modification is shown in Figure 2g–j. Compared to the DPV, the cross-section of the cell wall for acetylated DPV (ADPV) shows the significant fiber bulges and a great deal of micropores, which were sharply augmented with the increase of acetylation time. Especially for the ADPV for 5 h (ADPV-5), the amount and aperture of the micropores significantly increased, as shown in Figure 2j. During acetylation processes, the hydroxyl groups in the hemicellulose and amorphous regions of cellulose were gradually replaced by the acetyl groups [19]. As a hydrophobic group, the substitute of acetyl group would reduce the binding force of hydrogen bonds between celluloses and space out the cellulose fibers; accordingly, the fiber bulges and micropores appeared on the cross-sections of the cell wall [20], and this phenomenon became more distinctness with the increase of acetylation time. On the other hand, the contact angle test indicates that the initial contact angle of DPV was 0° due to its hydrophilia [21]; however, the initial contact angles for ADPV-1, -3, and -5 were gradually increased, and were 45°, 54°, and 59°, respectively; at the same time, the moisture absorption of all ADPVs are also evaluated, as shown in Figure 2f. The moisture absorption was gradually increased with the relative moisture varied from 25% to 100% for all specimens, noting that all ADPVs still displayed a lower moisture absorption than DPV, and the moisture absorption of ADPVs decreased with the increase in acetylation time. The significant increase in the initial contact angle and the decrease of moisture absorption prove that the hydrophobicity of the ADPV was enhanced due to the hydrophobic acetyl groups restraining the absorption of moisture [22].

The microstructure of the cross-sections of TDPV and TADPVs are displayed in Figure 3. Obviously, the epoxy resin was fully filled into the cell cavity and the intercellular spaces of DPV; however, a great deal of gaps between the cell wall and epoxy resin can be observed in Figure 3a, which means the interface connection was not so tight. It is worth noting that a few gaps could be found from TADPV-1, as shown in Figure 3b, which are mainly ascribed to the hydrophobic acetyl groups on the cell wall, enhancing the interface bonding with epoxy resin. Furthermore, the cross-section of TADPV-3 was extremely smooth and neat, and there were no gaps on the interface between cell wall and epoxy resin—only a few microvoids could be observed on the cell wall (Figure 3c). This is because the entering of a great deal of epoxy resin into cell wall resulted in a favorable interface compatibility. Nevertheless, TADPV-5 showed significant microvoids on its cross-section of the cell wall (Figure 3d). Based on the above microstructure of ADPV-5 and TADPV-5, the excessive acetylation treatment resulted in a lot of micropores that emerged on the cell wall of DPV, which loosened the cell wall structure. During epoxy resin polymerization, the cohesion of epoxy resin in the cell wall was greater than that of the cellulose fibers, resulting in a large number of microcracks in the cell wall, although a good interfacial compatibility was obtained [23].

### 3.2. Chemical Components and Crystal Structure Analysis

The chemical components and crystal structure of all specimens are displayed in Figure 4. The peaks of the FTIR spectrum of PV hat appeared at 3430 cm^−1^ and 2913 cm^−1^ are mainly attributed to the O–H and C–H stretching vibration of cellulose [24]. It is clear that these peaks are almost unchanged after delignification, as shown in Figure 4a. However, the peaks located at 1505 cm^−1^ and 1425 cm^−1^, which are due to the aromatic skeleton vibrations and the benzene ring-hydrogen bond stretching vibration of lignin [24], respectively, could not nearly be observed on the FTIR spectrum of DPV. Similarly, the characteristic peaks of lignin located at 1462 cm^−1^ and 1367 cm^−1^ completely disappeared. In addition, the relative content of the chemical components indicated that the lignin content substantially decreased from 22.69% to 0.41% after delignification, while the relative content of hemicellulose and cellulose was significantly enhanced (Figure 4b). This further indicates the removal of lignin.

Subsequently, Figure 4c demonstrates the change of chemical components of DPVs after acetylation modification. For the FTIR spectrum of DPV, the absorption peak at 1745 cm^−1^ is caused by the stretching vibration of C=O groups of hemicellulose, and the peak at 1370 cm^−1^ is due to the bending vibrations of the C-H groups of cellulose and hemicellulose, and the characteristic peak at 1243 cm^−1^ represents the stretching vibrations of C-O groups of cellulose [25,26]. After acetylation modification, the intensity for these peaks was markedly enhanced, because the O–H groups of cellulose and hemicellulose were substituted for CH_3_-C=O- (acetyl groups); meanwhile, the intensity of these absorptions was gradually increased with the extending of the acetylation time. These facts clearly indicate that acetyl groups were successfully grafted on the bulky DPV. Moreover, the substitution degrees of the acetyl groups were calculated [26], which were 0.135, 0.456, and 0.648 for ADPV-1, ADPV-3, and ADPV-5, respectively (Figure 4d). This substitution degree is significantly higher than those of acetylated woods with other catalysts, which suggests that 4-Dimethylpyridine shows a more efficient catalyst.

In order to further evaluate the effect of acetylation treatment on the crystal structure of ADPVs, the XRD curves are shown in Figure 4e, in which the diffraction peaks located at 15.8° and 22.6°, as well as a small diffraction peak at 35°, are attributed to the *I_101_*, *I_002_*, and *I_040_,* crystal planes of cellulose, respectively [27]. Significantly, these characteristic peaks did not change for all specimens, which still retained a typical cellulose I crystal structure. The crystallinity for all specimens was calculated according to the Segal method, as displayed in Figure 4f. The degree of crystallinity for DPV was evidently enhanced from 53.8% to 69.8% due to delignification; however, it is notable that the crystallinity of ADPVs had a slight decrease after acetylation treatment, and the maximum reduction for ADPV-5 was about 4%, which could be distributed to the destroying of hydrogen bonds in the crystalline regions caused by the substitution of acetyl groups.

### 3.3. Optical Performance and Flexibility of TADPVs

The optical performance and flexibility were investigated, as shown in Figure 5. Judging from the real products displayed in Figure 5a, TADPVs were more transparent than TDPVs. The light transmittance of all specimens is shown in Figure 5b under the spectrum range of 400~800 nm. It is distinct that the transmittance of TADPVs is significantly higher than that of TDPV in the whole spectrum. Among them, TADPV-3 had a highest light transmittance of 91% at a wavelength of 550 nm, which was greatly superior to TDPV, whose transmittance was only 78% (Figure 5c). Here, the enhancement of light transmittance was mainly caused by following reasons: (1) the introduce of acetyl groups on the cell wall enhanced the hydrophobicity of DPV, which resulted in a good interface compatibility between cell wall and epoxy resin in the cavities [28], and (2) the micropores appeared on the cross-sections of cell wall were completely filled with epoxy resin, forming a uniform and stable cell wall structure; accordingly, the light scattering was significantly decreased due to the reduce of gaps. However, the transmittance of TADPV-5 was slightly diminished compared to that of TADPV-3, which was about 84%. This is because the microcracks that appeared in the cell wall caused by the bigger cohesion of epoxy resin during polymerization increased the interior light scattering. At the same time, the haze of all specimens was displayed in Figure 5d, it is comforting that the haze of TADPVs was slightly enhanced due to the acetylation treatment. The TADPV-3 displayed the highest haze value of 74% at a wavelength of 550 nm, which was higher than that of TDPV, of which the haze was 62% (Figure 5e). In this research, the improvement of interfacial compatibility greatly decreased the light scattering in the surface and interior of TADPVs; accordingly, the haze was increased and the transparence was enhanced. This is advantageous to the optical materials with a higher haze and transmittance, such as transparent conductive substrates, for example a car windshield [29]. In addition, in order to estimate the effect of acetylation treatment on the flexibility of TADPVs, a typical bending test was implemented, in which the bending radius and bending angle were the key parameters for flexibility. The results indicated that the bending radius and bending angle parallel to the grain of TADPV-3 were 2 mm and 130°, respectively; however, the TDPV exhibited a higher bending radius and smaller bending angle, which were 2.25 mm and 118° respectively, as shown in Figure 5f. This exhibited that the flexibility of the TDPV was also improved due to the enhancement of the interfacial compatibility [30].

### 3.4. Mechanical Properties

The variation of the density of DPV after the acetylation and infiltration of epoxy resin is shown in Figure 6a. It is clear that the density of ADPV had little enhancement due to the substitution reaction of acetyl groups, whose molecular weight was more than the hydroxy groups of wood. Furthermore, the density of TDPV or TADPVs was increased by about three times against DPV or ADPVs due to the filling of the epoxy resin. It is a remarkable fact that the TADPVs exhibited a higher density than TDPV, which further proved that the increase of porosity caused by acetylation treatment provided more rooms for epoxy resin. The stress−strain curves and mechanical properties of all specimens are displayed in Figure 6b and Table 1, respectively. Although a lot of epoxy resin was filled in the DPV, the tensile strength and tensile modulus of TDPV were only 67 MPa and 112 MPa, respectively, which was much lower than that of PV. This is mainly attributed to the substantial remove of lignin, which weakened the bonding among the cellulose fibers; on the other hand, epoxy resin exhibited an extremely low tensile strength, around 1 MPa. Accordingly, TDPV had a lower mechanical performance. However, the mechanical properties of all TADPVs were significantly improved and the TADPV-3 displayed the highest mechanical properties, in which the tensile strength and tensile modulus were 102 MPa and 198 MPa, which increased by 52% and 64%, respectively. Significantly, these mechanical properties were higher than the plastic substrates used in flexible electron devices, such as polyethylene terephthalate (PET), polyethylene naphtholate (PEN), and polystyrene (PST) [31]. To more specifically explain the effect of interface compatibility on mechanical properties, the microfracture interfaces of all specimens were investigated. As shown in Figure 6c, it is clear that the damage of the interface was basically tensile failure of a single fiber, in which the wood fibers and epoxy resin were completely separated; accordingly, the TDPV presented on weak tensile strength. This is because the removal of lignin weakened the bonding force between wood fibers, while the epoxy resin in the PV was just a filling. However, this phenomenon is not observed in Figure 6d,e,f. The destruction forms of the interface for TADPV-1 was mainly ascribed to the fracture of columnar fiber bundles consisting of fibers and epoxy resin, which means a better interface compatibility was formed due to the acetylation modification. Notably, it is clear that the wood fiber was tightly enwrapped by epoxy resin and it formed a cross-linking composite for TADPV-3; therefore, its destruction form presents on a typical plane fracture, as shown in Figure 6e, which resulted in a superior mechanical performance. However, the fracture interface of TADPV-5 was not as neat as TADPV-3 and was presented on a great deal of peeled laminate structures because of the microcracks that appeared on the cell wall.

### 3.5. Thermal Conductivity Analysis

Superior thermal insulation is a key characteristic for transparent conductive substrates; here, the thermal conductivity of all specimens was investigated with a laser-infrared camera thermal-conductivity system. The steady-state temperature distribution was recorded using a FLIR Merlin MID Infrared (IR) camera. The temperature differences (ΔT) between the top and bottom surfaces of the TADPVs and the bulky epoxy resin were recorded [34], respectively, as shown in Figure 7a. Clearly, the thermal conductivities of all TADPVs had a small enhancement compared to TDPV, in which TADPV-3 demonstrated the highest thermal conductivity of 0.34 Wm^−1^ K^−1^, while the thermal conductivity of TDPV was only 0.30 Wm^−1^ K^−1^. This is mainly attributed to the increase of the infiltrated epoxy resin in TADPVs. The bulky epoxy resin possessed a higher thermal conductivity of 0.43, obtained from our testing results, which were higher than those of the fast-growing poplar [35], whose value was about 0.23 Wm^−1^ K^−1^. Accordingly, the TADPVs exhibited increased thermal conductivities. In spite of this, the thermal conductivity of the TADPV-3 was close to those of PET and PEN [36], whose thermal conductivities were 0.31 Wm^−1^ K^−1^ and 0.34 Wm^−1^ K^−1^, respectively, as displayed in Figure 7b. These results indicated that the TADPV-3 could be used as a candidate for he flexible transparent substrate of electron devices due to its excellent thermal insulation.

## 4. Conclusions

In this study, in order to prepare superior transparent fast-growing poplar veneers, we mainly studied the effect of modification with acetic anhydride using a green catalyst of DMAP on the interface of the bulk fast-growing poplar veneers after delignification. The results indicate that the interface hydrophobicity of PV was greatly enhanced due to the introduction of acetyl groups, which resulted in a more permeation of EP; therefore, the interface compatibility between PV and EP was significantly improved. The obtained transparent fast-growing poplar veneers displayed a superior optical performance and flexibility, outstanding mechanical properties, and lower coefficient of thermal conductivity, which were significantly better than those of the plastic substrates used in flexible electron devices. Therefore, the obtained TADPV could be as a candidate for a flexible transparent substrate of electron devices.

## Figures and Tables

**Figure 1 polymers-14-00257-f001:**
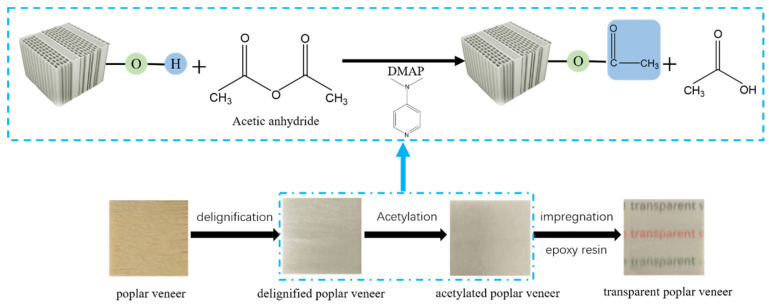
Scheme of the preparation process of transparent poplar veneers by acetylation modification.

**Figure 2 polymers-14-00257-f002:**
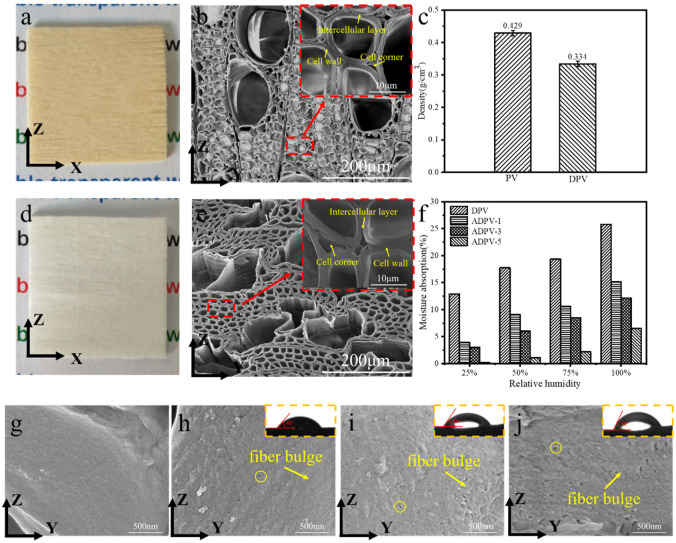
Microstructure and moisture absorption of all specimens for (**a**,**d**) the material objects of PV and DPV, respectively; (**b**,**e**) the microstructure of PV and DPV, respectively; (**c**) the density of PV and DPV; (**f**) moisture absorption of DPV and ADPV-1, -3, and -5; (**g**–**j**) the microstructure of DPV and ADPV-1, -3, and -5, respectively (X-longitudinal direction; Y-tangential direction; Z-radial direction of the wood).

**Figure 3 polymers-14-00257-f003:**
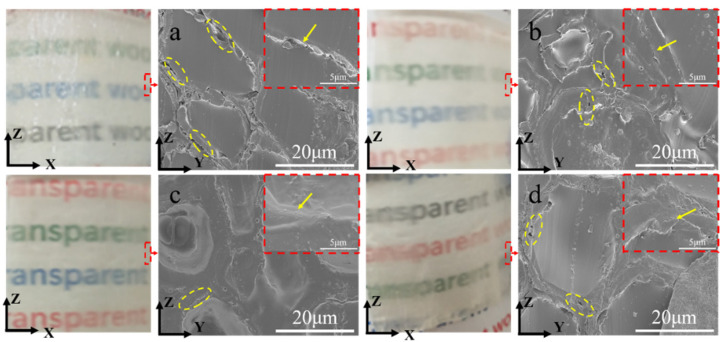
Microstructure of TDPV and TADPV-1, -3, and -5, named (**a**–**d**) respectively.

**Figure 4 polymers-14-00257-f004:**
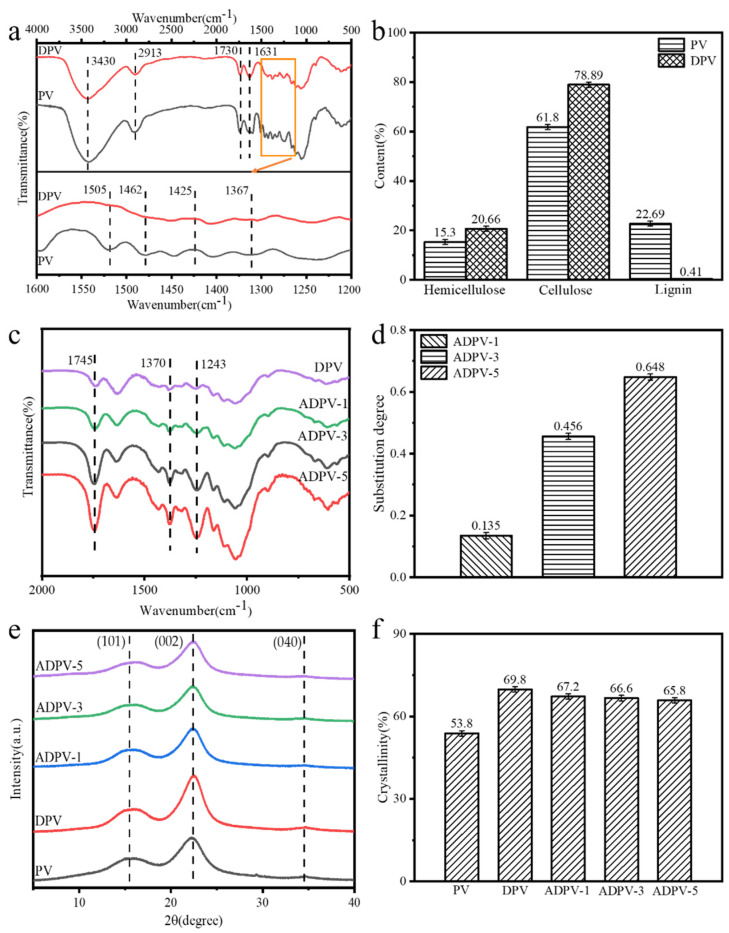
Chemical components and crystal structure of all specimens for (**a**) the FTIR spectra of PV during delignification, (**b**) the change of chemical components of PV after delignification, (**c**) the FTIR spectra of DPV after acetylation treatment, (**d**) the substitution degrees of acetyl groups for ADPVs, (**e**) the XRD curves of ADPVs, and (**f**) the degree of crystallinity for all specimens.

**Figure 5 polymers-14-00257-f005:**
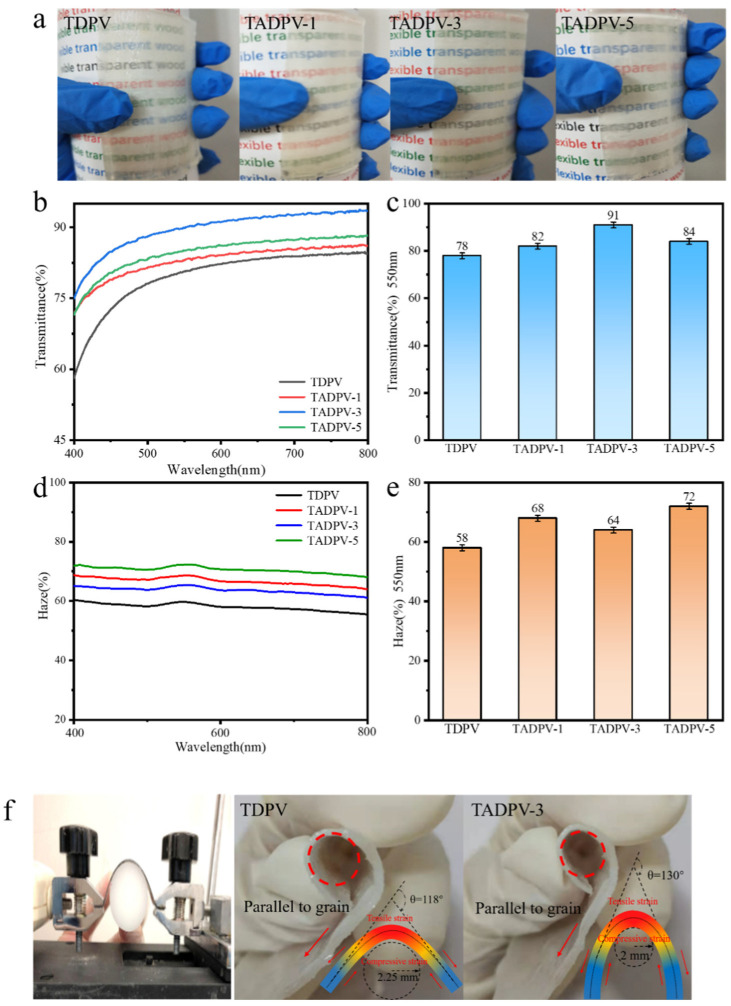
Optical performance and flexibility of TADPVs for (**a**) the real products; (**b**,**c**) the light transmittance of all specimens under 400–800 wavelengths and at a 550 nm, respectively; (**d**,**e**) the haze of all specimens under 400–800 wavelength sand at a 550 nm, respectively; and (**f**) the flexibility of TDPV and TADPV-3.

**Figure 6 polymers-14-00257-f006:**
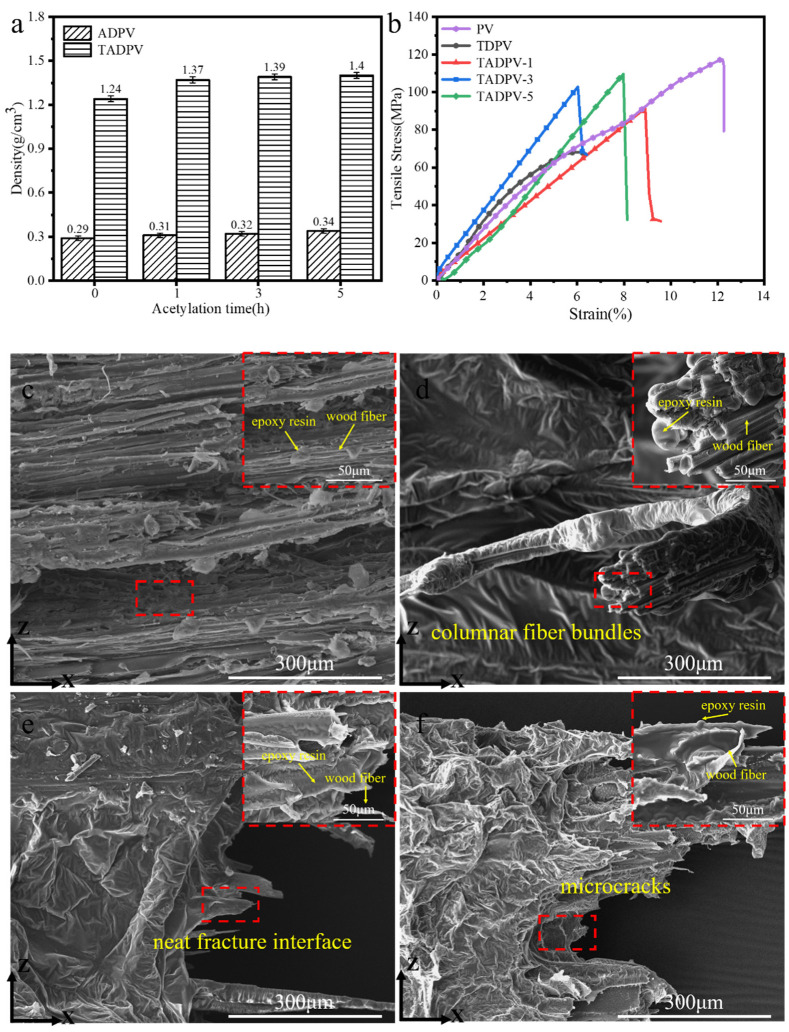
Variation of density and destruction forms of the interface of all specimens for (**a**) th variation of density; (**b**) stress−strain curves; (**c**–**f**) the destruction forms of the interface of TDPV, TADPV-1, TADPV-3, and TADPV-5, respectively.

**Figure 7 polymers-14-00257-f007:**
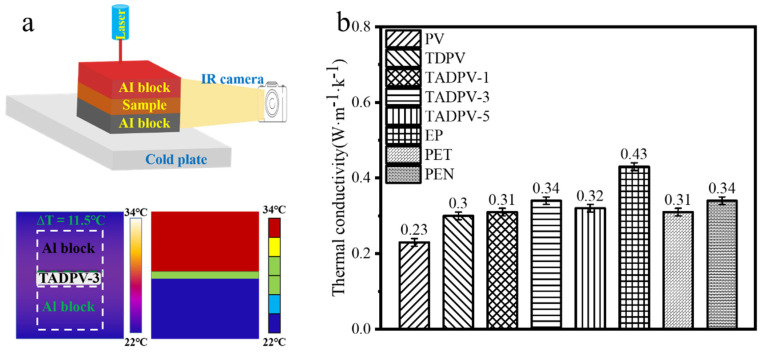
Thermal conductivity testing equipment and the coefficient of thermal conductivity of all specimens for the (**a**) thermal-conductivity measuring system, and (**b**) the comparison of all TADPVs and plastic substrates used in flexible electron devices.

**Table 1 polymers-14-00257-t001:** Comparison of the mechanical properties of all of the specimens and other plastic substrates.

Sample	Tensile Stress (MPa)	Tensile Modulus (MPa)	Elongation at Break (%)
TDPV	67	112.13	6.04
TADPV-1	91	106.37	8.91
TADPV-3	102	198.89	12.54
TADPV-5	101	127.54	7.91
EP	9.6	0.75	
PET	55 [32]	48	5.87
PEN	74 [33]	70	6.82

## Data Availability

Not applicable.

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
