# Peer review of "Preparation of Transparent Fast-Growing Poplar Veneers with a Superior Optical Performance, Excellent Mechanical Properties, and Thermal Insulation by Acetylation Modification Using a Green Catalyst"

_polymers, 2022, doi:10.3390/polym14020257_

Round 1

Reviewer 1 Report

Dear Authors,

It is a carefully and well-prepared article on the topic of biomaterials that can replace plastics in the future.

My comments are few and they concern minor corrections, including editing ones.

I present my comments in a synthetic form.

Keywords – line: 31
All keywords should be lowercase, e.g. fast-growing poplar veneers, acetylated ...

Materials - lines: 96-97

The term "fast-growing" contained in the title will be without coverage if there is no specific species or a hybrid of the tested poplars in subsection "Materials"

There are about 30 species of poplars in the world with different growth dynamics. There are also many hybrids and artificially created interspecies hybrids. In my opinion, it should be clearly indicated what type (species or hybrid) of wood was used with its Latin name. The term fast-growing poplar is imprecise.

Some poplar species produce heartwood.
Is the tested wood heartwood or sapwood?

The density of the wood is given, but the moisture content is not specified.
Is it the air-dry density, i.e. with a moisture content of 12%?

Figure 2 – lines: 183-186
In the caption under figure 2, an explanation of the X, Y, and Z axes should be provided:
X – longitudinal direction; Y – tangential direction, Z – radial direction of the wood

This is important because wood is an anisotropic material.

Lines: 358 – 360
There is no complete supplementary information required by the publisher (Polymers template). The following items were omitted: Author Contributions, Funding, Data Availability Statement.

References – lines: 363-460
The font size is too large for all references.

Yours sincerely
Reviewer

Author Response

Reviewer1: It is a carefully and well-prepared article on the topic of biomaterials that can replace plastics in the future. My comments are few and they concern minor corrections, including editing ones.I present my comments in a synthetic form.

Keywords – line: 31
All keywords should be lowercase, e.g. fast-growing poplar veneers, acetylated ...

We already modified them, please check it out.

Materials - lines: 96-97

The term "fast-growing" contained in the title will be without coverage if there is no specific species or a hybrid of the tested poplars in subsection "Materials"

There are about 30 species of poplars in the world with different growth dynamics. There are also many hybrids and artificially created interspecies hybrids. In my opinion, it should be clearly indicated what type (species or hybrid) of wood was used with its Latin name. The term fast-growing poplar is imprecise. Some poplar species produce heartwood. Is the tested wood heartwood or sapwood? The density of the wood is given, but the moisture content is not specified. Is it the air-dry density, i.e. with a moisture content of 12%?

According to your suggestion, we clearly indicated what type (species or hybrid) of wood and its Latin name, also the sapwood and moisture content was added to the content.

Figure 2 – lines: 183-186
In the caption under figure 2, an explanation of the X, Y, and Z axes should be provided:
X – longitudinal direction; Y – tangential direction, Z – radial direction of the wood. This is important because wood is an anisotropic material.

We already added the content of X – longitudinal direction; Y – tangential direction, Z – radial direction to the line 189.

Lines: 358 – 360
There is no complete supplementary information required by the publisher (Polymers template). The following items were omitted: Author Contributions, Funding, Data Availability Statement.

 We have done the supplementary information to the manuscript.

References – lines: 363-460
The font size is too large for all references.

We already modified them.

Reviewer 2 Report

An excellent written, presented and discussed article. The session 3.1. is really outstanding. Well done to authors.

My only suggestion is a slight modidfication of the article, so please the term modification, since the term acetylation is adequate. Alternatively, you can omit the term acetylation and write modification with acetic anhydride.

Author Response

Reviewer 2: My only suggestion is a slight modidfication of the article, so please the term modification, since the term acetylation is adequate. Alternatively, you can omit the term acetylation and write modification with acetic anhydride.

Thank you for the suggestion, we already made a slight modification using the modification with acetic anhydride.